# Vibration Characteristics of a Dual-Rotor System with Non-Concentricity

**Shengliang Hou [1], Lei Hou [1,2,\*], Shiwei Dun [3], Yufeng Cai [3], Yang Yang [2] and Yushu Chen [1]**

[1] School of Astronautics, Harbin Institute of Technology, Harbin 150001, China; 17B918014@stu.hit.edu.cn (S.H.); yschen@hit.edu.cn (Y.C.)

[2] Applied Mechanics and Structure Safety Key Laboratory of Sichuan Province, School of Mechanics and Engineering, Southwest Jiaotong University, Chengdu 610031, China; 181042yy@163.com

[3] Factory of Xiang Yang Hang Tai Power Machinery, Xiangyang 441000, China; dsw39397850@163.com (S.D.); caiyufeng51888@163.com (Y.C.)

\* Correspondence: houlei@hit.edu.cn

**Abstract:** A finite element model of an aero-engine dual-rotor system with intermediate bearing supported by six bearings is set up. Three modes of non-concentricity caused by the assembly process are defined, namely parallel non-concentricity, front deflection angle non-concentricity and rear deflection angle non-concentricity. The influence of the non-concentricity on the vibration characteristics of the dual-rotor system is investigated in detail. The results show that the parallel non-concentricity and the front deflection angle non-concentricity have a significant influence on the bending vibration modals of the high-pressure rotor and the low-pressure rotor, but have little influence on the local vibration modals of the rotors. With the increase in the magnitude of the non-concentricity, the natural frequencies of the bending modals decrease continuously, and the mode shapes of bending modals and that of local modals may be interchanged, leading to the emergence of bending modals in advance. Therefore, the key parameters to be controlled in the assembly process are the parallel non-concentricity and the front deflection angle non-concentricity. In order to prevent the bending modal of the dual-rotor system from appearing in advance, it is necessary to control the parallel non-concentricity within 2 mm and the front deflection angle non-concentricity amount within 0.18°.

**Keywords:** dual-rotor system; non-concentricity in the assembly process; finite element model; vibration characteristics

## 1. Introduction

Due to the manufacture and installation errors of the support components and the deformation of elastic support structure, the concentricity of high-pressure and low-pressure rotors, which may cause excessive vibration, is hardly guaranteed during the assembly process of the dual-rotor system. The extreme example will cause the damage of the turboshaft, couplings and other components. Thus, it will affect the flight safety of the aircraft. Whether the multi rotor "series" rotor system formed by coupling or the multi rotor "parallel" rotor system formed by coupling of intermediate bearings, there exist many types of misalignment and non-concentricity, no matter what the formation of the multi rotors is [1].

For the type of series-multi rotors represented by power turbine, the research subjects mainly focus on the misalignment of the coupling and the supporting. Xia et al. [2] introduced the misalignment mechanism of the coupling in detail, pointing out that the torque of misalignment will be generated in the misalignment of coupling, which will lead to the transmission of the torque of misalignment to the rotor system. Then, the rotor system will produce more complex vibrational phenomena. The research on the misalignment of all kinds of couplings showed that the misalignment of the gear coupling



will lead to a negative value of the angular meshing stiffness, resulting in the instability of the rotor system [3]. Several harmonic frequencies of torsional vibration excited by the misalignment of coupling, which is mainly composed of 1X, 2X and 3X components, are observed [4]. However, the misalignment fault cannot be determined by the doubling frequency component only. In the locked state, the phase difference in the same direction perpendicular to the axis on both sides of the coupling is also an important distinguishing condition for judging the misalignment type of the gear coupling [5]. The multiple doubling frequencies of the rotor system is caused by the misalignment moment of hook coupling angle [6]. Misalignment of the flange coupling can lead to an odd number of doubling frequencies [7]. Supporting misalignment mainly refers to the misalignment caused by bearing manufacturing, installation, and other reasons, among which the research on the misalignment of sliding bearing is relatively sufficient. Huber et al. [8] proposed that if the journal and the bearing bush axis are not aligned, the bearing edge load would be concentrated and the operation will be unstable. According to the characteristics of long necked sliding bearing, Arumugam [9] obtained that the misalignment of bearing can reduce the oil film thickness, and increase the friction and damping of system. Park [10] put forward that the change of misalignment angle has a direct impact on the film thickness and pressure distribution.

For the parallel multi rotor system, which is represented by the dual-rotor system, the research work mainly focuses on the misalignment of the coupling and the non-concentricity between the rotors. Han et al. [11] put forward six modes for the advanced turbofan engine rotor system structural features, namely, the rotor system supporting non-concentricity of the fan section, the sleeve gear coupling misalignment of the low-pressure rotor system, the supporting non-concentricity of the high-pressure rotor system, the supporting non-concentricity of a high-pressure rotor system, the non-concentricity of inner and outer rotors, and so on. They summarized that the rotor system would produce lateral and axial vibration caused by the sleeve gear coupling misalignment and the supporting non-concentricity of the roller bearing. There would be frequency conversion and frequency doubling components, but no proportional relationship with the value of misalignment. A basic dual-rotor system contains gas generator rotor and power turbine rotor. Both are connected by coupling which will cause the misalignment of the dual-rotors in the form of deflection angle by the coupling misalignment. For instance, the rigid gear sleeve coupling is a typical component that causes the misalignment of the rotor system. Ma et al. [12] conducted modeling and calculation of the sleeve gear coupling. Finding that the stiffness of the coupling would change, and stability of the coupling would be lost when the coupling is misaligned.

The non-concentricity may affect the dynamic characteristics of the rotor system and the fatigue failure of the bearing. Wu et al. [13] transformed the non-concentricity of supporting into the clearance model of bearing, and the non-concentricity of supporting was one of the important reasons leading to the fatigue failure of rolling bearing. Based on the flexible rotor system with multiple supports, Zhang et al. [14] obtained that the dynamic effect of the non-concentricity of the bearing varies with the change of the structural characteristics of the rotor, it may bring additional excitation frequency to the system. Xu et al. [15] studied the influence of the high-pressure rotor supporting non-concentricity on the dynamic characteristics of the double rotor system, which indicated that the non-concentricity would create a new resonance region in the amplitude frequency response of the system. Li et al. [16] established a dynamic model of a dual-rotor with intermediate support, which indicated that the vibration characteristics of the low-pressure rotor would be transmitted to high-pressure rotors. Lu et al. [17] established a dynamic model for a dual-rotor-bearing system with flexible coupling misalignment faults in the low-pressure rotor system. The results show that the vibration characteristics of misalignment faults in the low-pressure rotor can be transmitted to the high-pressure rotor and can be used to diagnose the misalignment faults. By numerical calculation, Feng et al. [18] showed that the non-concentricity of the support bearing may lead to the occurrence of 2X frequency components in high and

low-pressure rotors. Liu et al. [19] established the non-concentricity mechanical model of the multi span flexible rotor system under the non-concentricity excitation, which indicated that the non-concentricity would bring additional unbalanced excitation to the rotor system. Li et al. [20] established a dynamic model for an offset-disc rotor system with a mechanical gear coupling, which takes into consideration the nonlinear restoring force of rotor support and the effect of coupling misalignment. It was found that the conditions leading to the instability of periodic solution are the period doubling bifurcation and the secondary Hopf bifurcation. Song [21] studied the vibration mechanism of the two rotors non-concentricity, and pointed out that the non-concentricity fault of the high-pressure rotor is the dominant fault feature of the non-concentricity. Abdou [22] studied the influence of rotor misalignment on stability of finite width radial plain bearing, it is concluded here that increasing the value of misalignment degree yields to an increase in the value of critical bearing stability limit for a given value of misalignment direction angle. The critical stability number increased as the misalignment direction angle decreased and/or the steady state eccentricity ratio increased. Santo [23] studied the effect of rotor–tower interaction, tilt angle, and yaw misalignment on the aero elasticity of a large horizontal axis wind turbine with composite blades. Sinha [24] proposed a method to estimate the unbalance and misalignment of a flexible rotating machine from a single operation. Chen et al. [25] proposed a method to suppress the vibration of the multi-stage rotor, a simplified four-stage high-pressure rotor system was assembled according to the optimal assembly angles calculated in the simulations.

In this paper, a dual-rotor system of an aero-engine is taken as the research object to study the influence of the non-concentricity induced by the assembly process on the vibration characteristics of the system. Three kinds of non-concentricity modes are investigated, namely parallel non-concentricity, front angle non-concentricity and rear angle non-concentricity. Based on the three-dimensional solid finite element model, the influence of the non-concentricity magnitude on the vibration characteristics of the dual-rotor system under the three kinds of non-concentricity modes are investigated in detail. The reasonable control suggestions for the non-concentricity in the assembly process are given without changing the order of the vibration modals of the system.

## 2. FE Modeling and Experimental Set-Up of the Dual-Rotor System

### 2.1. The Model of the Dual-Rotor System

In this paper, the dual-rotor system of an aero engine is taken as the research object, as shown in Figure 1. It consists of a high-pressure rotor, a low-pressure rotor, coupling, bearing and other structures. It includes six supporting points, in which the No. 1, No. 2 and No. 6 bearings belong to low-pressure rotor bearings; the No. 1 and No. 2 bearings are fixed between the outer casing and the low-pressure rotor; the No. 6 bearing is fixed between the inner casing and the low-pressure rotor; the No. 3 and No. 5 bearings belong to the high-pressure rotor and are fixed between the inner casing and the high-pressure rotor; and the No. 4 bearing is the intermediate bearing, which is fixed between the high-pressure and low-pressure rotors.

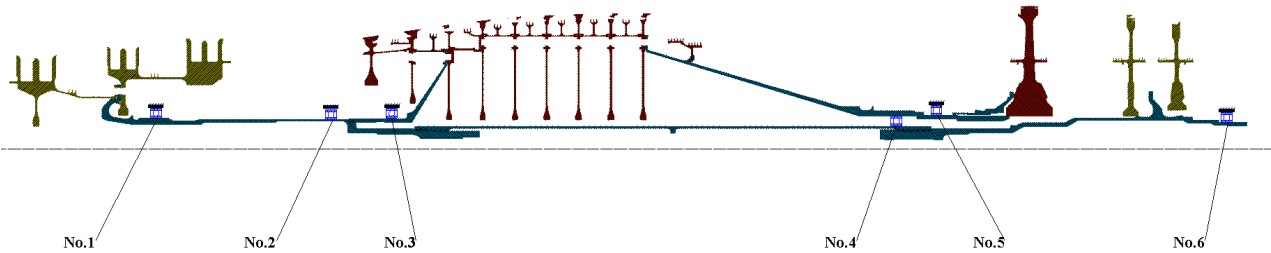

**Figure 1.** Supporting structure of dual-rotor system.

The actual structure is simplified, the bolt connection and coupling connection are simplified into adhesive structure; the position of tenon, chamfer and comb disc are

equivalent; the blade is treated by the method of equivalent mass center, and the solid finite element model is established [26]. Among them, solid element solid185 is used for the high-pressure rotor, low-pressure rotor and gearbox, and the support structure is simplified as linear spring element combin14. According to the empirical calculation formula, the linearization results of the stiffness parameters of the support structure are obtained. The parameter results are added to the finite element linear element structure, as shown in Figure 2.

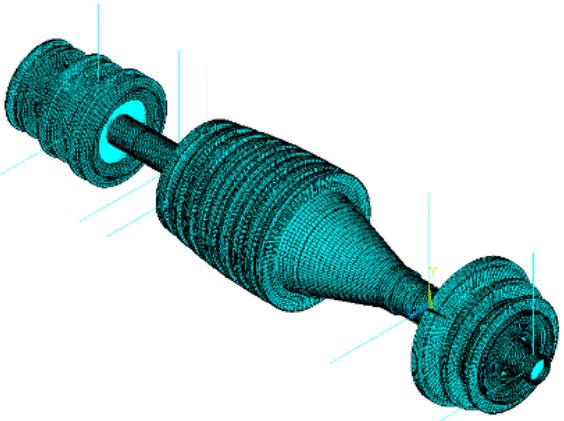

**Figure 2.** Finite element model of dual-rotor system.

The model includes high-pressure rotor, low-pressure rotor, fulcrum bearing, intermediate bearing, gearbox and other structures, totaling 582,814 elements and 9,000,000 nodes. The low-pressure rotor includes low-pressure compressor disk, low-pressure compressor shaft, low-pressure turbine disk and low-pressure turbine shaft, totaling 48,964 elements and 83,046 nodes. The high-pressure rotor includes high-pressure compressor disk, high-pressure compressor shaft and high-pressure compressor shaft. There are 47,328 units and 76,731 nodes in the structure of pressure turbine disk and high-pressure turbine shaft, 486,103 units and 8,840,227 nodes in the structure of gearbox and accessories.

## 2.2. Experimental Set Up of the Dual-Rotor System

Taking a dual-rotor aero-engine as the experimental object, two motors are used to drive the high- and low-pressure rotors to rotate. Its structure is shown in Figure 3.

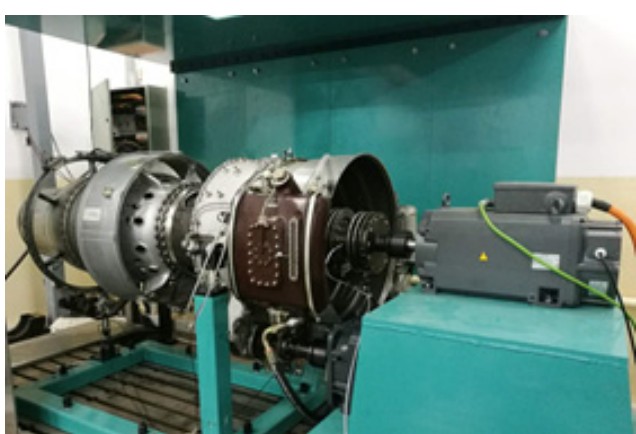

**Figure 3.** Aero Engine Test Bench.

The acceleration sensors are set on the aero-engine. The natural frequency and mode of the dual-rotor system are measured by the hammer method. The testing principle of its vibration characteristics is shown in Figure 4. The acceleration sensors are arranged

on the dual-rotor system of the aero-engine. Three measuring points are arranged on the low-pressure compressor, two measuring points on the high-pressure compressor, and one measuring point is arranged at the rear support of the low-pressure rotor. The natural frequency and modal tests are completed by determining the hammer hitting point on the rotor structure of the engine.

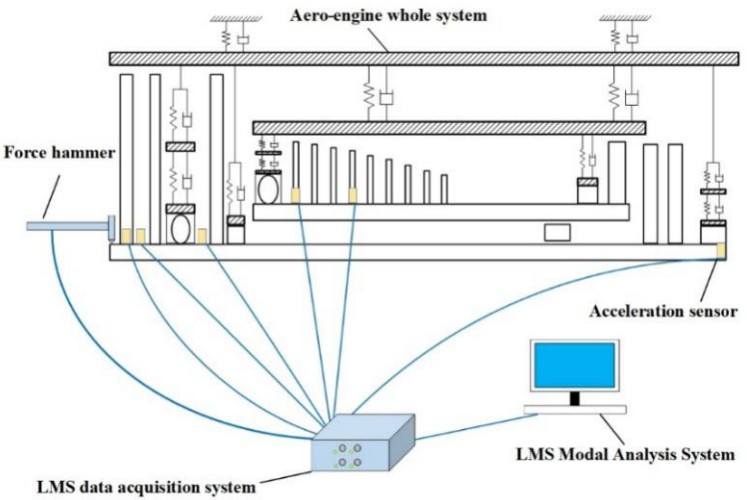

**Figure 4.** Schematic diagram of dual-rotor modal test for aero-engine whole system.

An LMS system was used in the testing process, and the test equipment used is shown in Figure 5.

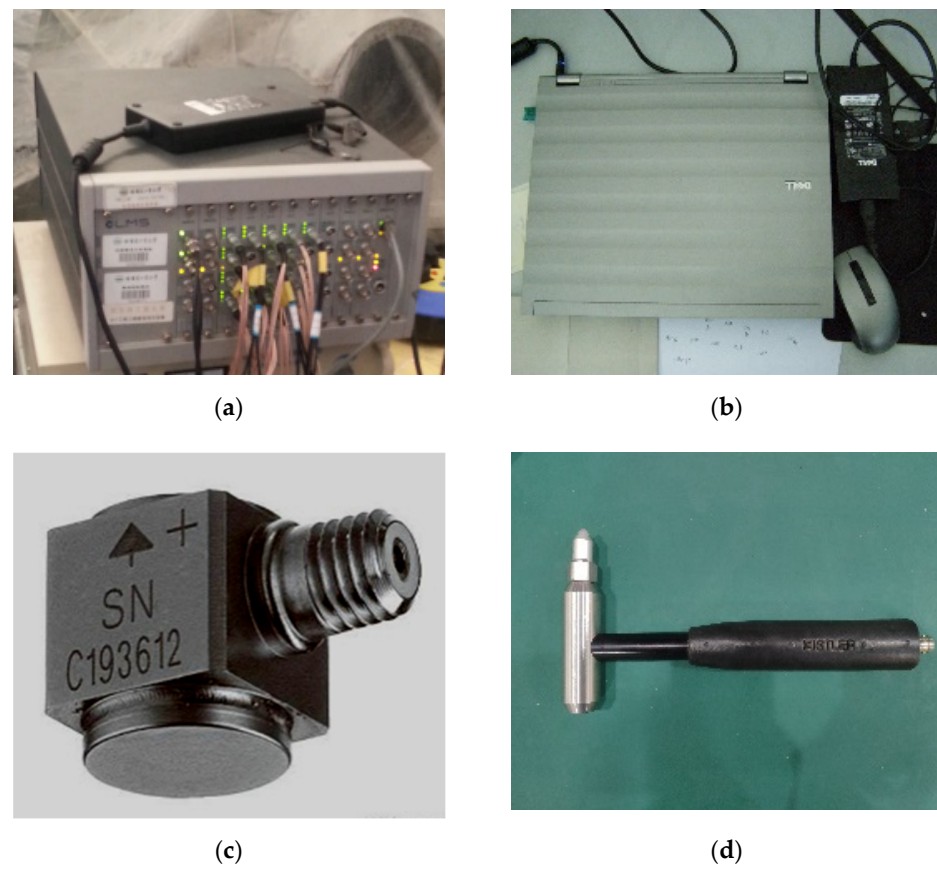

**Figure 5.** LMS modal testing and analysis equipment: (**a**) LMS data acquisition device; (**b**) LMS modal analysis system; (**c**) KISTLER acceleration sensor; (**d**) Force hammer.

The measurement points of the acceleration sensor on the aero-engine rotor system are shown in Figure 6.

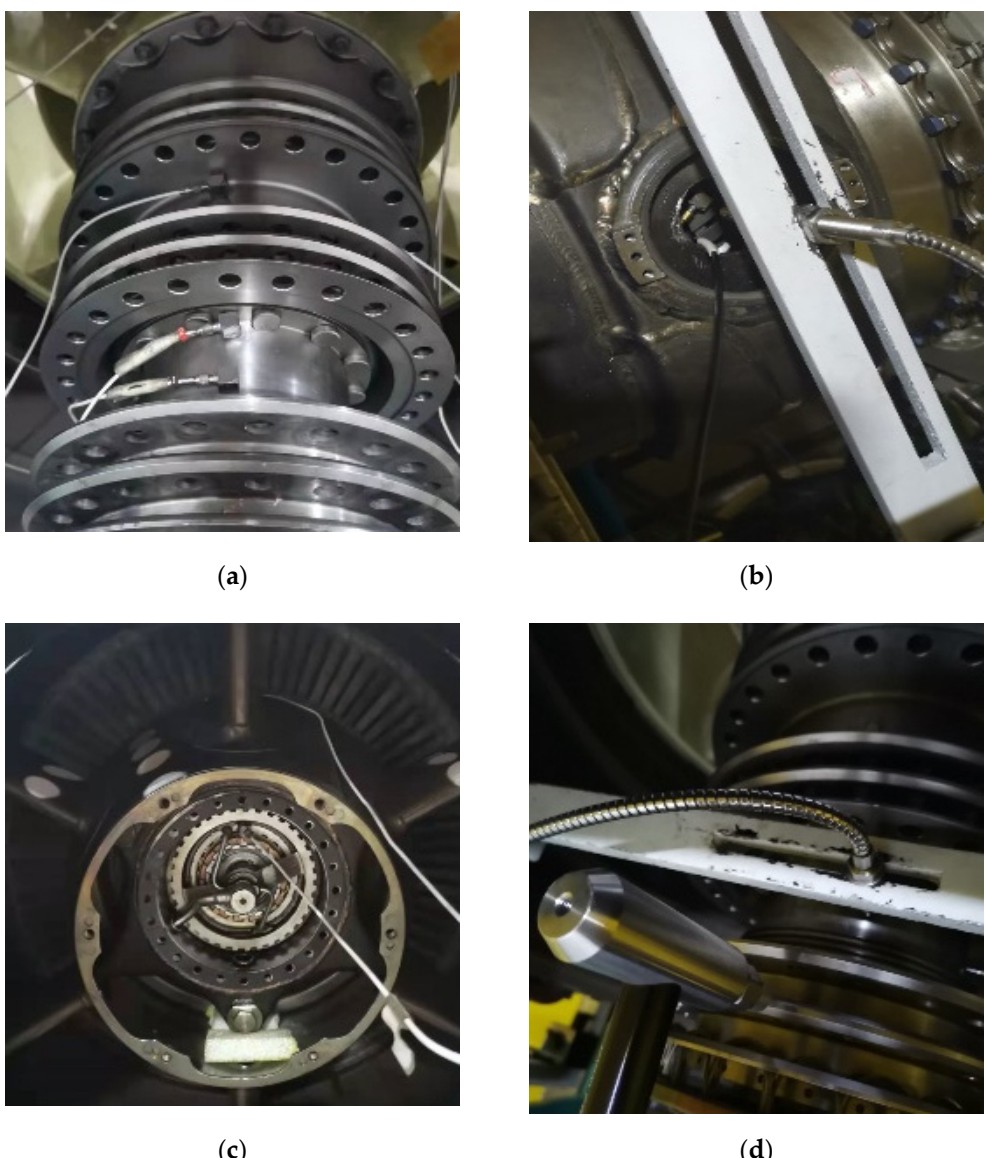

(**a**)　　　　　　　　　　　　　　　　(**b**)

(**c**)　　　　　　　　　　　　　　　　(**d**)

**Figure 6.** Sensor measuring point arrangement for dual-rotor system of aero-engine: (**a**) Measuring points of the low-pressure compressor; (**b**) Measuring points of the high-pressure compressor; (**c**) Measuring points of the rear supporting of the low-pressure turbine; (**d**) Measuring points of the force hammer.

LMS system is used to establish a simplified beam element model. The high- and low-pressure rotors are simplified into linear beam elements and connected by two intermediate bearings. The model structure is shown in Figure 7.

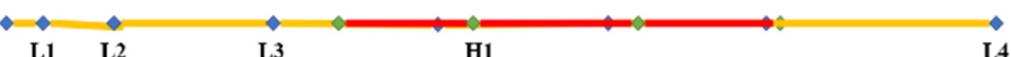

**Figure 7.** Beam element model established by LMS (L1, L2, L3 and L4 are low-pressure rotor measuring points, H1 is high-pressure rotor measuring points, yellow line is low-pressure rotor, red line is high-pressure rotor).

Taking the second-order modal shape as an example, the comparison between the modal shapes obtained by testing and the simulation results is shown in Figure 8.

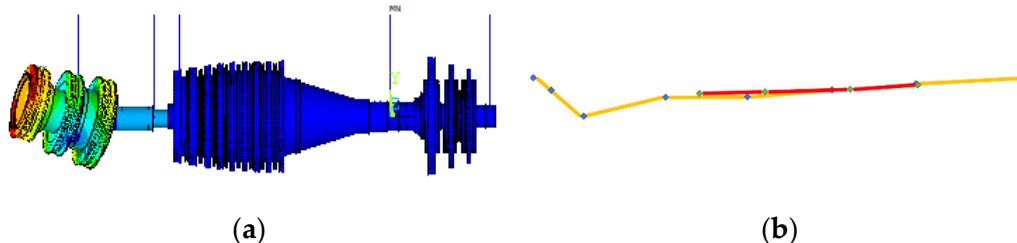

(**a**)                    (**b**)

**Figure 8.** The second-order modal shape comparisons between experiments and FEM analyses: (**a**) The second-order modal shape of FEM analyses; (**b**) The second-order modal shape of LMS.

Table 1 shows the comparison of the FEM analysis of the model. The relative errors of the FEM analyses and the first three natural frequencies of the experimental results are obtained. The minimum relative error of the first order natural frequency is 6.9%, and the maximum relative error of the third order is 16.7%. Through comparison, it is found that the relative error between experiments and FEM analysis is small, which proves the accuracy of the model.

**Table 1.** Natural frequencies comparisons between experiments and FEM analyses.

| Order | Experiment (Hz) | Calculate (Hz) | Error |
|---|---|---|---|
| First | 123.1 | 114.6 | 6.9% |
| Second | 193.4 | 210.1 | 8.6% |
| Third | 307.6 | 256.2 | 16.7% |

## 3. The Definition of Three Modes of Non-Concentricity

### 3.1. The Cause of Misalignment and Non-Concentricity

There are many types of non-concentricity of a dual-rotor system. The fault tree of the possible causes is shown in Figure 9.

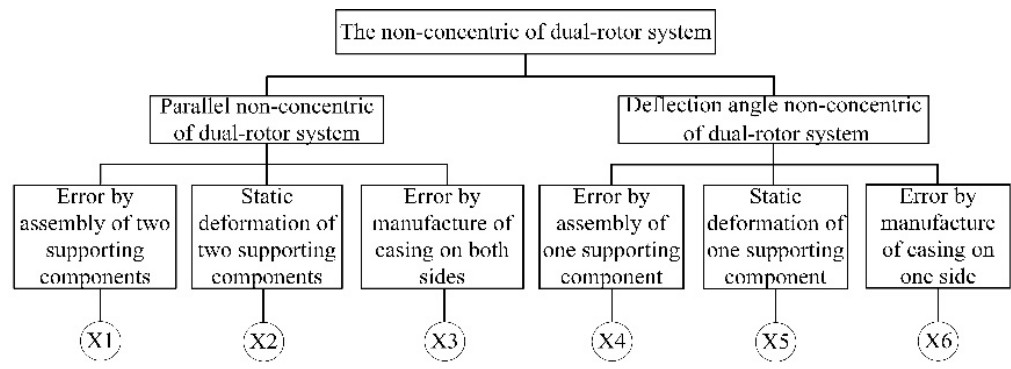

**Figure 9.** Non-concentricity fault tree of dual-rotor systems.

The misalignment in rotor systems can be classified into two types: parallel misalignment and angle misalignment. The causes of the parallel misalignment are manufacturing error, installation error and static deformation of the elastic supports on both sides of the rotor, while the causes of the angle misalignment are manufacturing error, installation error, and static deformation of the one side elastic support.

From the perspective of the dual-rotor system structure described in previous section, the No. 3 support position of the high-pressure rotor is an elastic support. The No. 4 bearing is an intermediate bearing. Coupled with the No. 5 bearing, it can also be regarded as an

elastic support. In addition to the manufacturing errors of support structures, due to the deformation of the elastic supports, the elastic support of the front and rear support points of the high-pressure rotor is prone to static deformation, which leads to the decentraction of the front and rear support points of the high-pressure rotor. The concentricity of the axis of the two rotors cannot be accurately guaranteed, and the problem of the concentricity of the two rotors is prone to occur in the assembly process. If there is an error at one support of the dual-rotor system, it will result in angle misalignment. If there are errors at two supports, it will result in parallel misalignment.

### 3.2. Three Modes of the Non-Concentricity

The non-concentricity in dual-rotor systems can be categorized into three modes. The first mode is that the high-pressure rotor axis produces parallel decentraction with the low-pressure rotor axis to form parallel non-concentricity, as shown in Figure 10, the second mode is that the high-pressure rotor produces decentraction with the low-pressure rotor axis to form front deflection angle non-concentricity, as shown in Figure 11. The third mode is just opposite to the second mode, as shown in Figure 12.

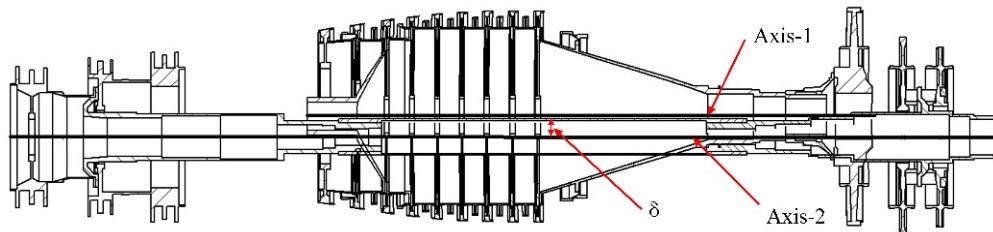

**Figure 10.** Parallel non-concentricity of dual-rotor.

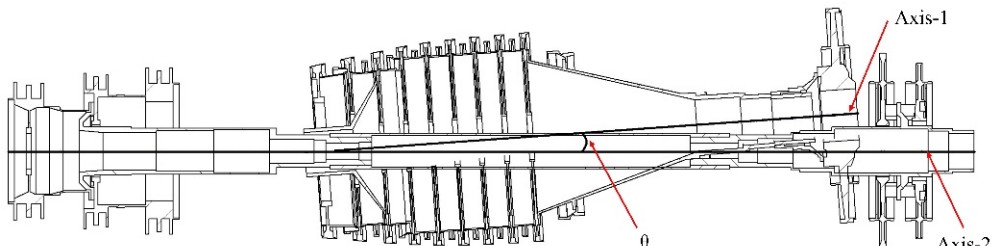

**Figure 11.** Front deflection angle non-concentricity of dual-rotor.

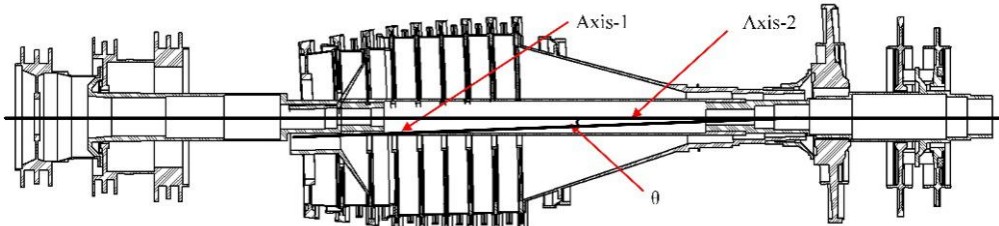

**Figure 12.** Rear deflection angle non-concentricity of dual-rotor.

The main reason for parallel non-concentricity of the dual-rotor is that the elastic supports at the front and rear support of the high-pressure rotor have static de-formation at the same time or the support components have manufacturing and installation errors at the same time. As shown in Figure 10, Axis 1 is the axis of the high-pressure rotor and Axis 2 is the axis of the low-pressure rotor. δ refers to the decentraction formed by the parallel non-concentricity of the dual-rotor.

The main reason for the front deflection angle of the dual-rotor is that the rear support of the high-pressure rotor has static deformation or the support component have manufac-

turing and installation errors. As shown in Figure 11, Axis 1 is the axis of the high-pressure rotor, Axis 2 is the axis of the low-pressure rotor, and θ is the angle formed by the front deflection angle non-concentricity of the dual-rotor.

The main reason for the front deflection angle of the dual-rotor is that the elastic supports at the front support of the high-pressure rotor have static deformation or the support component have manufacturing and installation errors. As shown in Figure 12, Axis 1 is the axis of the high-pressure rotor, Axis 2 is the axis of the low-pressure rotor, and θ is the angle formed by the rear deflection angle non-concentricity of the dual-rotor.

### 3.3. Realization of the Non-Concentricity in the Dual-Rotor System Model

In the finite element model, the high-pressure rotor is translated upward to produce parallel decentraction, forming a model of parallel non-concentricity. By controlling the displacement of the high-pressure rotor, the parallel non-concentricity of two axes is realized. In order to make the front deflection angle non-concentricity, the rear support of the high-pressure rotor is shifted upward to form the front deflection angle model of the dual-rotor. The front deflection angle non-concentricity is shifted upward by controlling the rear support of the high-pressure rotor. The front support of the high-pressure rotor is deflected upward to make the rear deflection angle, forming a model of the rear deflection angle non-concentricity.

## 4. Influence of Non-Concentricity on Natural Frequency and Mode Shape of the Dual-Rotor System

### 4.1. Natural Frequency and Mode Shape of the Dual-Rotor System without Non-Concentricity

For the established finite element model of the dual-rotor system, the natural frequencies and mode shapes are calculated. The first five modes and corresponding natural frequencies are obtained, which are listed in Table 2. The first modal is the local modal of the high-pressure turbine; the second modal is the local modal of stage 1 disc of the low-pressure turbofan rotor; the third modal is the overall bending modal of the high-pressure rotor; the fourth modal is the local modal of stage 2 and stage 3 discs of the low-pressure turbofan rotor; the fifth modal is the overall bending modal of the low-pressure rotor.

**Table 2.** The first five modals.

| Order | Natural Frequency (Hz) | Mode Shape |
|:---:|:---:|:---:|
| 1st | 114.6 | Local vibration mode of HP Turbine |
| 2nd | 210.1 | Local vibration mode of LP Compressor |
| 3rd | 256.2 | Bending vibration mode of HP Rotor |
| 4th | 284.7 | Local vibration mode of LP Compressor |
| 5th | 327.8 | Bending vibration mode of LP Rotor |

### 4.2. The Influence of the Parallel Non-Concentricity on the Natural Frequency and Modal Shape of the Dual-Rotor System

In this subsection, δ is selected in the range of 0 mm ~ 3 mm. Based on same rule, the first five natural frequencies and modal shapes of the dual-rotor with parallel non-concentricity are calculated. The change curves of the first five natural frequencies are given out in Figure 13. Curves with different colors correspond to different orders of natural frequency In Figure 13, the horizontal axis represents the different values of parallel non-concentricity of the dual-rotor system. The vertical axis represents the corresponding frequency of the dual-rotor system with set value parallel non-concentricity. With the increase in δ, the 1st, 2nd and 4th orders of natural frequency do not change. While the 3rd and 5th orders decrease with the increase in the δ. Thus, at points A and B, the frequency orders of 2 and 3 and 4 and 5 interchange.

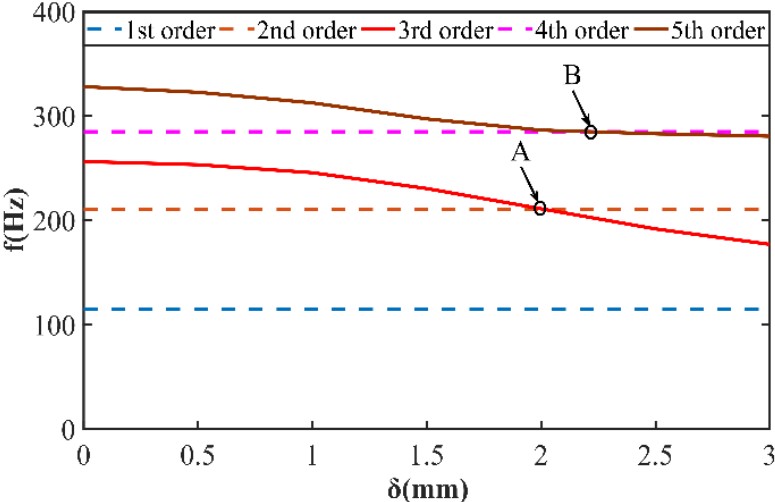

**Figure 13.** Variation trend of natural frequency of parallel non-concentricity in the range of 0 mm~3 mm.

More in detail, when δ increases from 0 to 3 mm, the 1st, 2nd, and 4th order natural frequencies change little, while the 3rd and 5th order natural frequencies change obviously. With the increase in δ, the 3rd and 5th order natural frequencies decrease. When δ increases to 2.1 mm, the 3rd order natural frequency will be less than the 2nd order natural frequency. Thusm the corresponding 2nd and 3rd order modals interchange. When δ continues to increase to 2.25 mm, the 5th natural frequency will be less than the 4th natural frequency, and the corresponding 4th and 5th modals will be interchanged.

In order to further explore the change of the 3rd order bending modal and 5th order bending modal under the parallel non-concentricity, the 3rd and 5th order natural frequencies of the system are extracted, Following Equation (1), the change rates can be calculated. Selecting 0.5 mm, 1 mm, 1.5 mm, 2 mm, 2.5 mm and 3 mm, the change rates of the 3rd order are obtained. The calculation results are shown in Table 3.

$$\omega = (f_i - f_0)/f_0, \tag{1}$$

where $f_0$ represents the natural frequency of the system when there is no parallel offset. $f_i$ represents the natural frequency under the different parallel non-concentricity. $\omega$ represents the change rate of natural frequency.

**Table 3.** The 3rd order natural frequency error under different parallel non-concentricity of the 3rd order.

| Decentraction (mm) | Natural Frequency (Hz) | Error |
|---|---|---|
| 0 | 256.17 | 0 |
| 0.5 | 253.11 | 1.2% |
| 1 | 245.45 | 4.2% |
| 1.5 | 230.28 | 10.1% |
| 2 | 210.92 | 17.7% |
| 2.5 | 191.61 | 25.2% |
| 3 | 176.84 | 31% |

According to the analysis of the results listed in Table 3, the change amount and change rate of the natural frequency of the bending modal of the 3rd order increase with the increase in the parallel non-concentricity. When the parallel non-concentricity reaches 3 mm, the change rate of the natural frequency reaches the maximum value of 31%.

The 3rd order modals with different parallel non-concentricity are extracted. The comparison of mode shapes is shown in Figure 14. The results show that with the increase in parallel non-concentricity, the 3rd order mode of the dual-rotor changes its shape from local vibration mode of the low-pressure turbofan to the bending vibration mode of the low-

pressure rotor. The deformation of the high-pressure rotor decreases with the increase in parallel non-concentricity, while the bending level of the low-pressure rotor increases with the increase in parallel non-concentricity. The maximum deformation position is gradually transferred from the high-pressure rotor to the low-pressure rotor, and the deformation of the transformed modal is larger and more obvious.

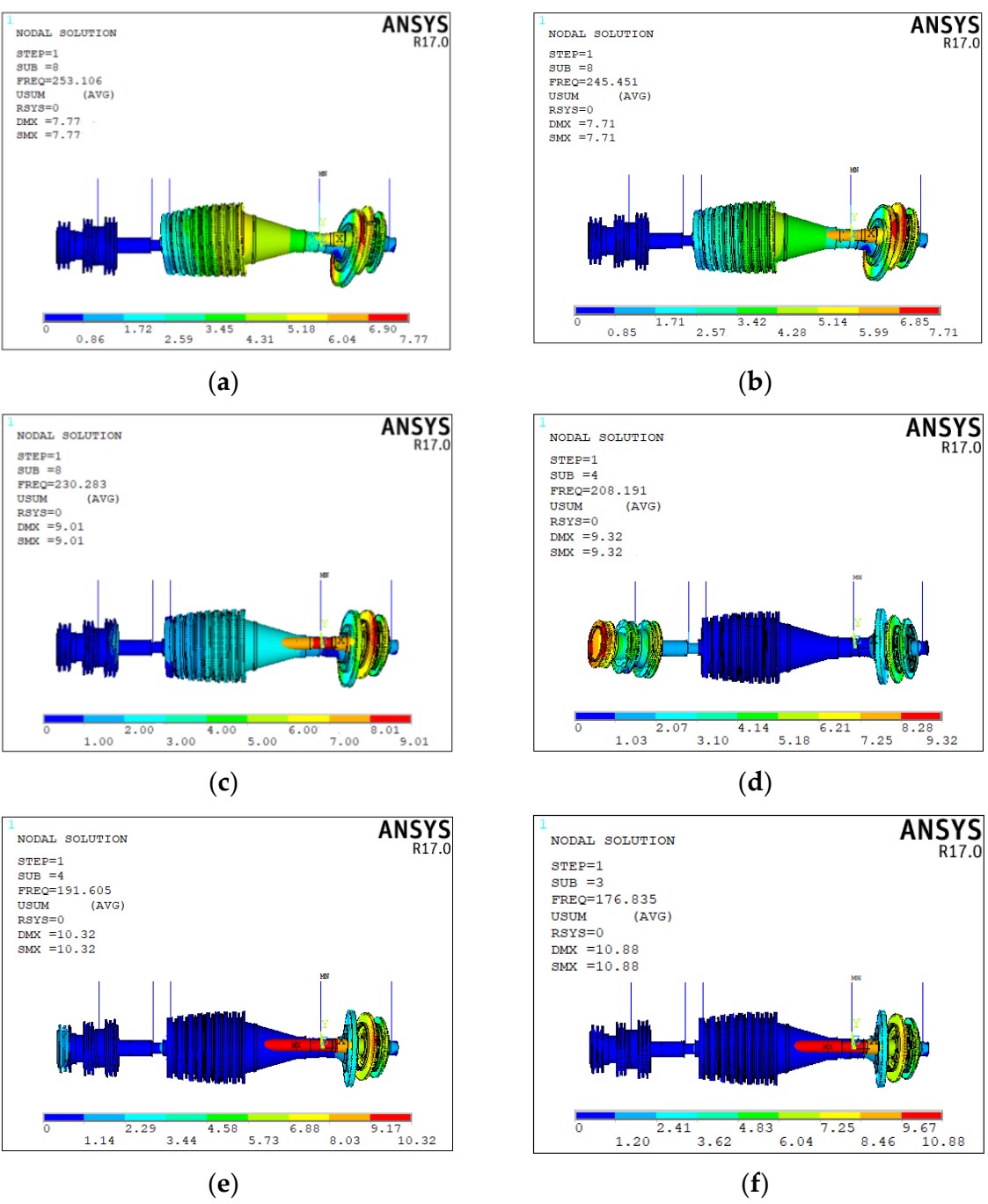

**Figure 14.** The 3rd order modals for contrast in different parallel non-concentricity: (**a**) δ = 0.5 mm; (**b**) δ = 1.0 mm; (**c**) δ = 1.5 mm; (**d**) δ = 2.0 mm (**e**) δ = 2.5 mm; (**f**) δ = 3.0 mm.

Similarly, we chose 0.5 mm, 1 mm, 1.5 mm, 2 mm, 2.5 mm, and 3 mm as the representative parameters to extract the bending vibration mode and its corresponding natural frequency of the 5th order. The change rates of natural frequencies with different decentration values are calculated and listed in Table 4. With the increase in parallel

non-concentricity, the change rate of natural frequency increases gradually. When the parallel non-concentricity reaches 3 mm, the change rate of natural frequency reaches the maximum value of 14.5%.

**Table 4.** The 5th order natural frequency error under different parallel non-concentricity of the 3rd order.

| Decentraction (mm) | Natural Frequency (Hz) | Error |
|:---:|:---:|:---:|
| 0 | 327.84 | 0 |
| 0.5 | 322.48 | 1.6% |
| 1 | 312.30 | 4.7% |
| 1.5 | 296.95 | 9.4% |
| 2 | 286.19 | 12.7% |
| 2.5 | 282.74 | 13.8% |
| 3 | 280.40 | 14.5% |

The comparison of 5th modal shapes with different decentration is shown in Figure 15. It is found that the 5th modal of the dual-rotor system changes from the bending modal of the low-pressure rotor to the bending modal of the high-pressure rotor with the increase in the parallel non-concentricity. This kind of bending modal is similar to the bending modal of the 3rd order high-pressure rotor, but there are some differences. It is different from the bending modal of the dual-rotor caused by the bending of the high-pressure rotor, it is the reverse bending of the high and low-pressure rotors. Additionally, with the increase in the parallel non-concentricity, the reverse bending of the two rotors becomes more and more severe. Compared with the bending modal of the 3rd order high-pressure rotor, this kind of bending modal has a more serious and destructive effect on the rotor.

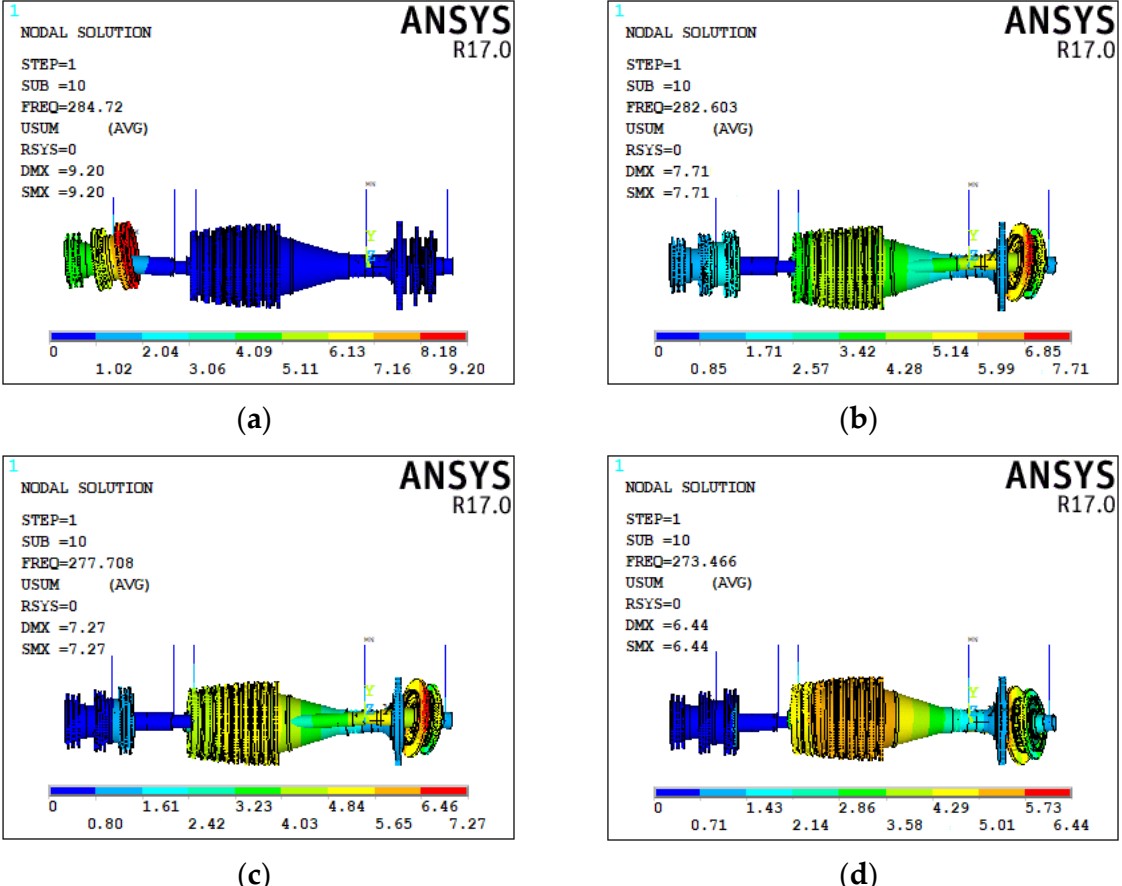

**Figure 15.** *Cont.*

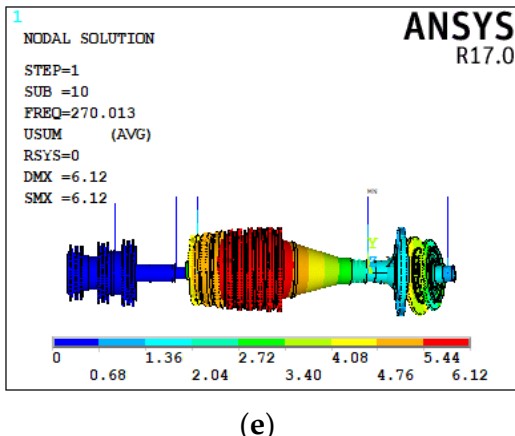
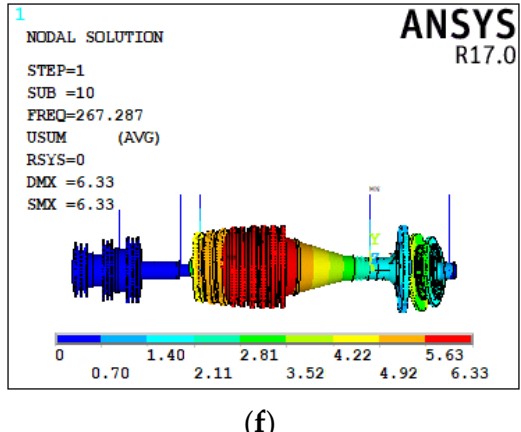

(**e**)                                                            (**f**)

**Figure 15.** The 5th order modals for contrast in different parallel non-concentricity: (**a**) δ = 0.5 mm; (**b**) δ = 1.0 mm; (**c**) δ = 1.5 mm; (**d**) δ = 2.0 mm; (**e**) δ = 2.5 mm; (**f**) δ = 3.0 mm.

*4.3. The Influence of the Front Deflection Angle Non-Concentricity on the Natural Frequency and Modal Shape of the Dual-Rotor System*

Following same analysis method, the angle non-concentricity is selected within the range of 0–0.26° to investigate the effects on the dual-rotor system. The corresponding relationship between the angle non-concentricity and the parallel non-concentricity can be got by Equation (2).

$$\delta = L\sin\theta \tag{2}$$

*L* is the distance from the front supporting of the high-pressure to the intermediate bearing.

Taking the angle of 0.02° as the spacing, the value corresponding relationship between $\theta$ and $\delta$ is found out. The results are shown in Table 5. Selecting the angle values of 0.05°, 0.1°, 0.15°, 0.2°, 0.25° and 0.3°, the natural frequency curves of each order with respect of angle value are shown in Figure 16. In Figure 16, the horizontal axis represents the angle of the front deflection angle non-concentricity, and the vertical axis represents the corresponding frequency. A is the modal interchange point of orders 2 and 3; B is the modal interchange point of orders 4 and 5.

In the case of the front deflection angle, non-concentricity of the dual-rotor system exists, and the natural frequencies of the 1st, 2nd, and 4th orders have not changed significantly, while the natural frequencies of the 3rd and 5th orders have changed significantly.

The rule is that the natural frequencies of the 3rd and 5th orders appear to decrease, as the angle of the front deflection angle non-concentricity increases. When it increases to 0.18°, the natural frequency of the 3rd order will be smaller than that of the 2nd order. The corresponding 2nd and 3rd mode shapes will be interchanged. At this time, $\delta$ is approximately 2.1 mm, which agrees with the parallel non-concentricity of the dual-rotor. When the angle value continues to increase to 0.22°, the 5th order natural frequency will be smaller than the 4th order natural frequency, and the 4th and 5th order modals will be interchanged. The 2nd and 3rd order modal interchange position is met earlier than the 4th and 5th order modal interchange position. Calculating the natural frequency change rates of the bending vibration mode of the 3rd order of the high-pressure rotor with different front deflection angle non-concentricity, the calculation results are shown in Table 6. As the angle value of the front declination angle non-concentricity increases, the change rate of the natural frequency gradually increases accordingly. When the front deflection angle reaches 0.3°, the natural frequency change rate reaches a maximum of 33.8%.

**Table 5.** The relationship between $\theta$ and $\delta$.

| $\theta$ (°) | $\delta$ (mm) | $\delta$ (mm) | $\theta$ (°) |
|---|---|---|---|
| 0 | 0 | 0.16 | 1.874 |
| 0.02 | 0.234 | 0.18 | 2.108 |
| 0.04 | 0.468 | 0.20 | 2.342 |
| 0.06 | 0.703 | 0.22 | 2.576 |
| 0.08 | 0.937 | 0.24 | 2.811 |
| 0.10 | 1.171 | 0.26 | 3.045 |
| 0.12 | 1.405 | 0.28 | 3.279 |
| 0.14 | 1.640 | 0.30 | 3.513 |

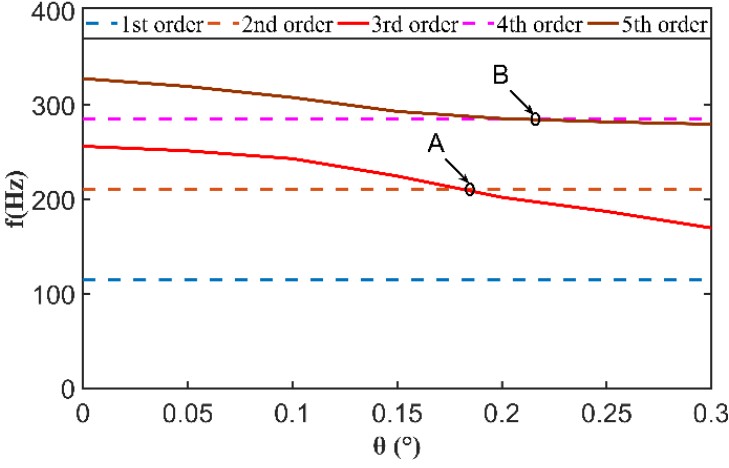

**Figure 16.** Variation trend of natural frequency of front deflection angle non-concentricity in the range of 0–0.3°.

**Table 6.** The change rate of the natural frequency of the 3rd order.

| Decentraction (°) | Natural Frequency (Hz) | Error |
|---|---|---|
| 0 | 256.17 | 0 |
| 0.05 | 251.32 | 1.9% |
| 0.1 | 243 | 5.1% |
| 0.15 | 224.60 | 12.3% |
| 0.2 | 202.06 | 21.1% |
| 0.25 | 187.14 | 27% |
| 0.3 | 169.64 | 33.8% |

With each set angle value of the front deflection angle non-concentricity, the 3rd order modal is extracted. As shown in Figure 17, the modals with different angle values are compared and analyzed. When the front deflection angle non-concentricity increases, the modal shape change from the bending of the high-pressure rotor to the bending of the low-pressure rotor. The maximum deformation position is shifted from the high-pressure rotor to the low-pressure rotor. When the angle increases, the amount of deformation of the low-pressure rotor gradually increases.

Similarly, 0.05°, 0.1°, 0.15°, 0.2°, 0.25°, and 0.3° are selected as representatives to extract the 5th order modals and their corresponding natural frequencies. The change rates are shown in Table 7. With the increase in the front deflection angle non-concentricity, the change rate of the natural frequency gradually increases. When the front deflection angle non-concentricity reaches 3 mm, the natural frequency change rate reaches the maximum of 14.7%.

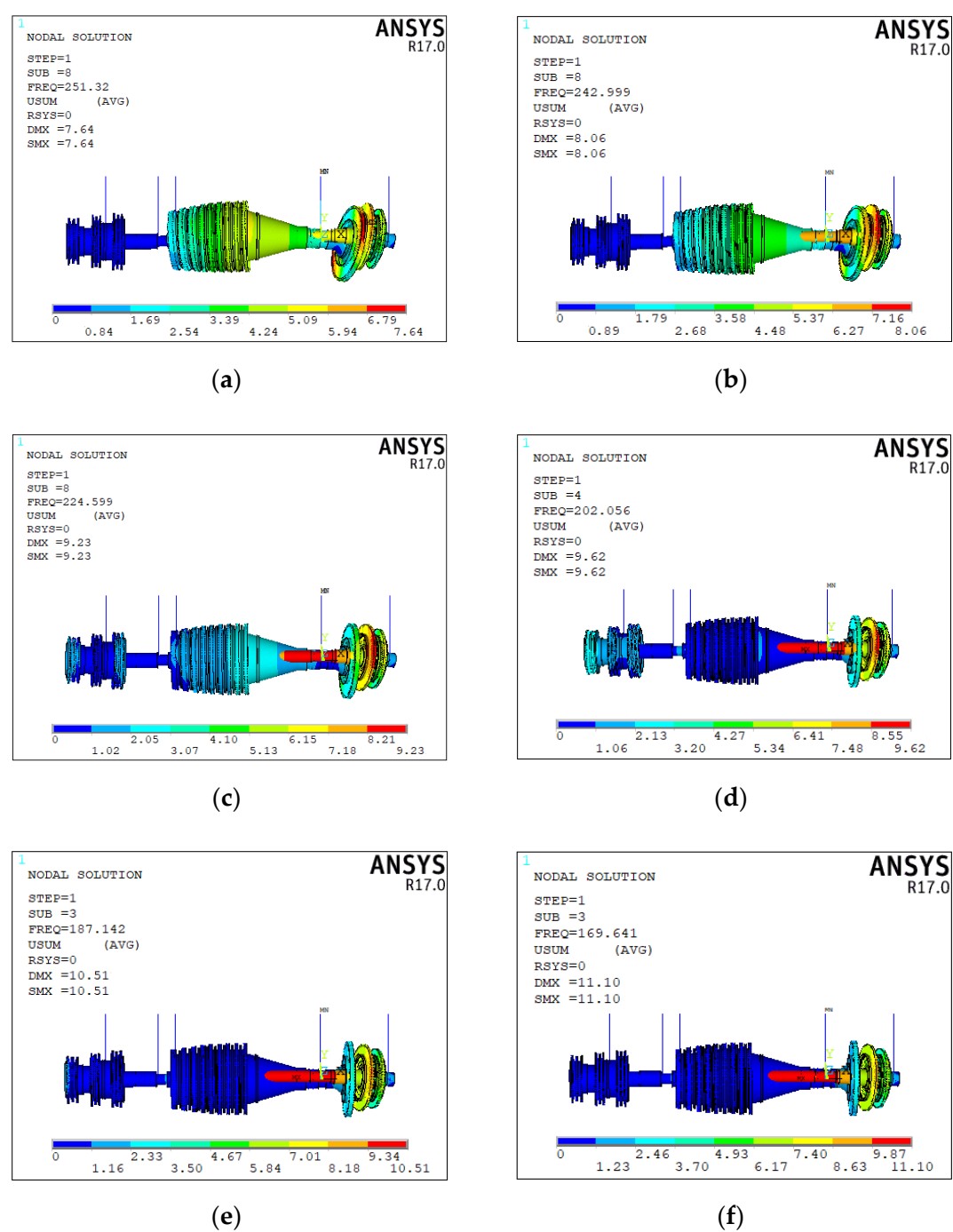

**Figure 17.** The 3rd order modal for contrast in different front deflection angle non-concentricity: (**a**) θ = 0.05°; (**b**) θ = 0.1°; (**c**) θ = 0.15°; (**d**) θ = 0.2°; (**e**) θ = 0.25°; (**f**) θ = 0.3°.

**Table 7.** The change rate of the natural frequency of the 5th order.

| Decentraction (°) | Natural Frequency (Hz) | Error |
|---|---|---|
| 0 | 327.84 | 0 |
| 0.05 | 319.54 | 2.53% |
| 0.1 | 307.82 | 6.11% |
| 0.15 | 293.03 | 10.6% |
| 0.2 | 285.52 | 12.9% |
| 0.25 | 281.98 | 14% |
| 0.3 | 279.55 | 14.7% |

The comparison of the 5th order vibration modes is shown in Figure 18. It finds that the change pattern of vibration mode is similar with the parallel non-concentricity. As the parallel non-concentricity increases, the 5th order vibration mode of the dual-rotor system changes from the bending modal of the low-pressure rotor to the bending modal of the high-pressure rotor, which is different from the bending modal of the 3rd order.

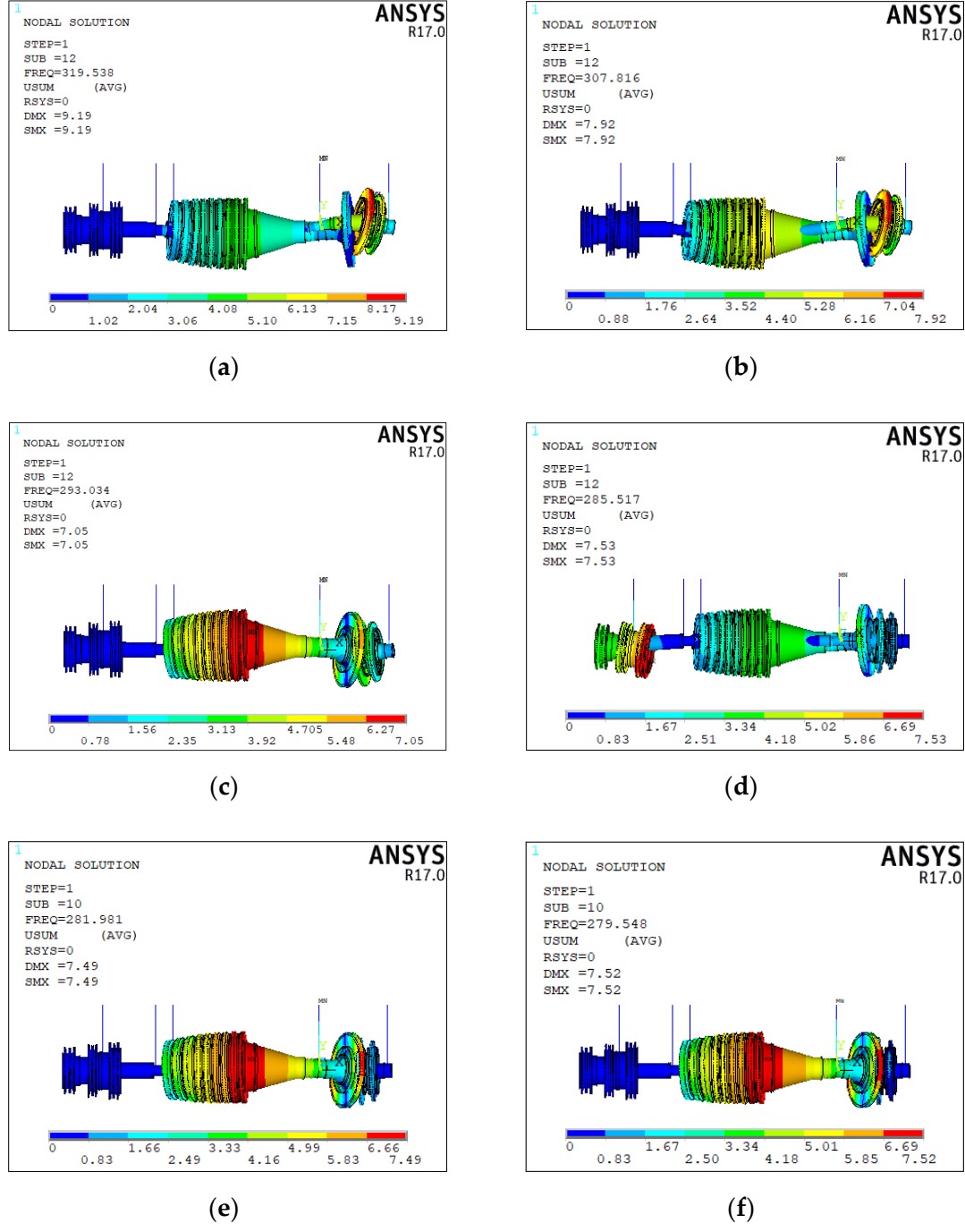

**Figure 18.** The 5th order modal for contrast in different front deflection angle non-concentricity: (**a**) θ = 0.05°; (**b**) θ = 0.1°; (**c**) θ = 0.15°; (**d**) θ = 0.2°; (**e**) θ = 0.25°; (**f**) θ = 0.3°.

### 4.4. The Influence of the Rear Deflection Angle Non-Concentricity on the Natural Frequency and Modal Shape of the Dual-Rotor System

Comparing the first five order natural frequencies with different rear deflection angle non-concentricity in the range of 0°–0.3°, the comparison results are shown in Figure 19. The natural frequencies of the first five modals did not change significantly.

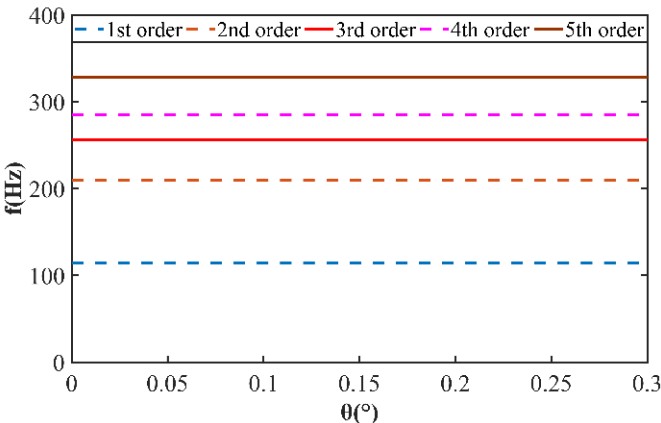

**Figure 19.** Variation trend of natural frequency of rear deflection angle non-concentricity in the range of 0~0.3°.

## 5. Influence of Non-Concentricity on Critical Speed and Suggestions

### 5.1. Influence of Non-Concentricity on Critical Speed

Based on the finite element model of the dual-rotor system, the critical speeds are calculated. The Campbell diagram of the dual-rotor system is shown in Figure 20. In order to investigate the effects of the non-concentricity, based on the two non-concentricity models of the dual-rotor system, the Campbell diagrams of the non-concentricity model are drawn and the critical speeds of the two non-concentricity models are calculated. The laws of parallel non-concentricity and front deflection angle non-concentricity on the critical speed are obtained. The comparison of parallel non-concentricity with values of 1 mm and 2 mm of the dual-rotor system is shown in Figure 21. Additionally, the comparison of 0.1°and 0.18° front deflection angle of the dual-rotor system is shown in Figure 22. Through the comparison, it is found that both parallel non concentricity and angle non concentricity could reduce the second-order critical speed and the third-order critical speed. However, both have little effect on the first-order critical speed.

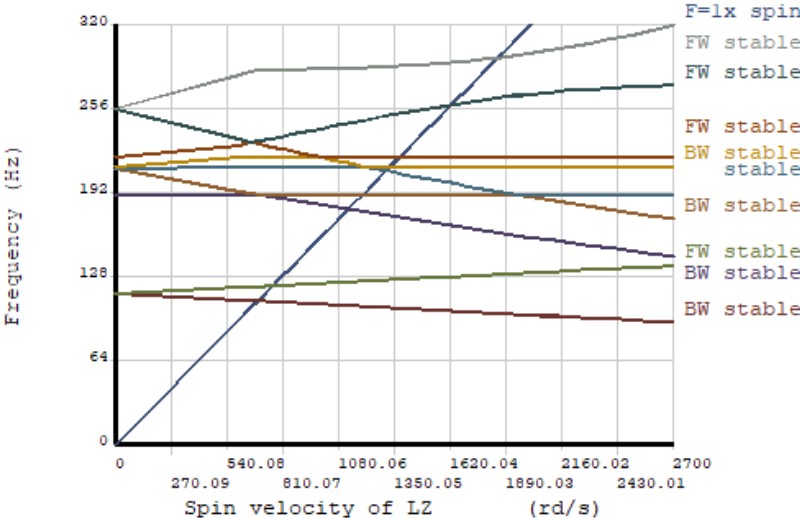

**Figure 20.** Campbell diagram of the dual-rotor system.

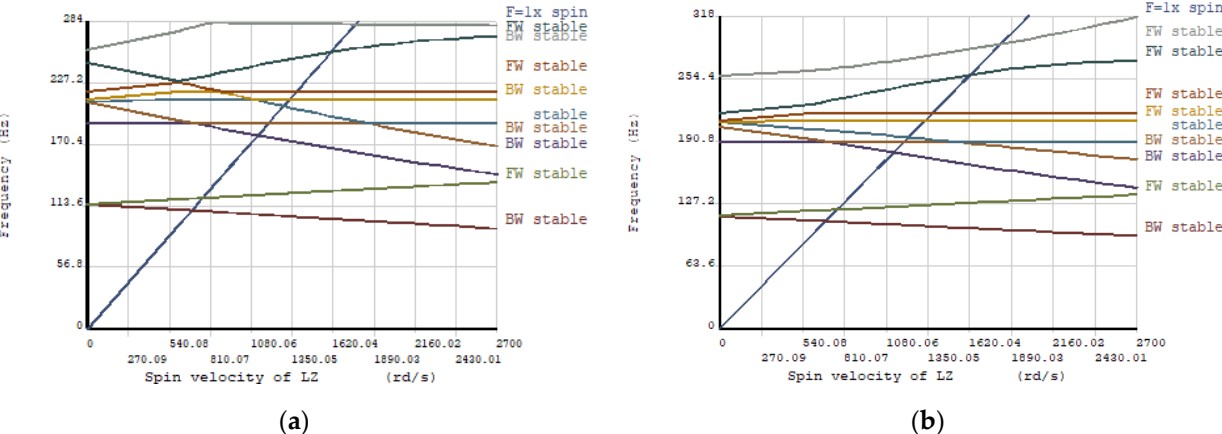

**Figure 21.** Campbell diagram of parallel non-concentricity: (**a**) δ = 1 mm; (**b**) δ = 2 mm.

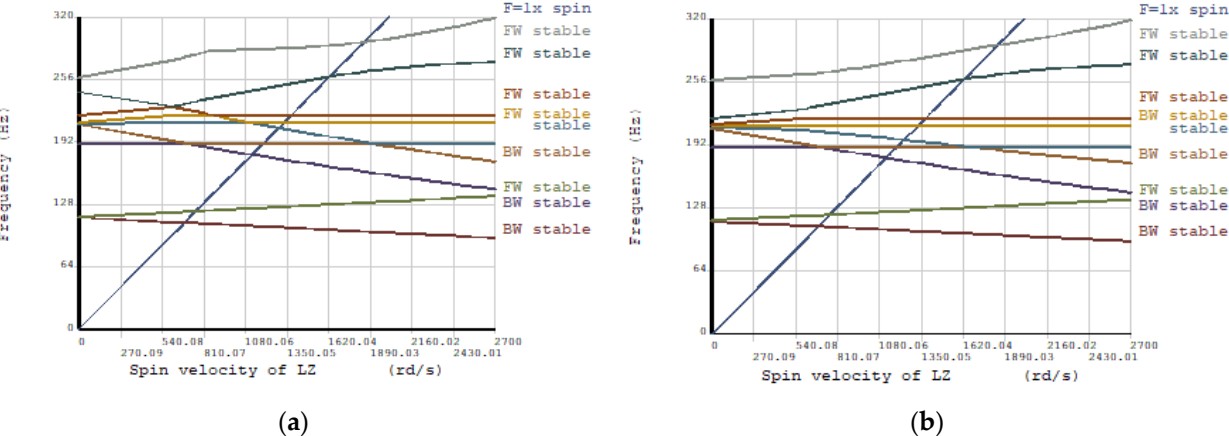

**Figure 22.** Campbell diagram of front deflection angle non-concentricity: (**a**) θ = 0.1°; (**b**) θ = 0.18°.

*5.2. Suggestions for Eccentric Control of Dual-Rotor Systems*

Based on the study of three different types of non-concentricity, it is worth noting that the rule of vibration characteristics of the parallel non-concentricity and the front deflection angle non-concentricity are similar. When the bending modal of the high-pressure rotor changes into the bending modal of the low-pressure rotor, the parallel non-concentricity of the dual-rotor is 2 mm, and the front declination angle non-concentricity is 0.18°. The corresponding modal comparison is given in Figure 23.

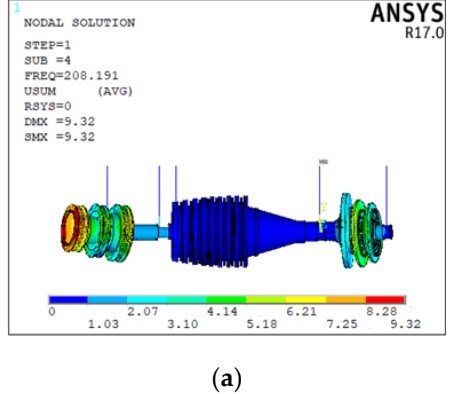 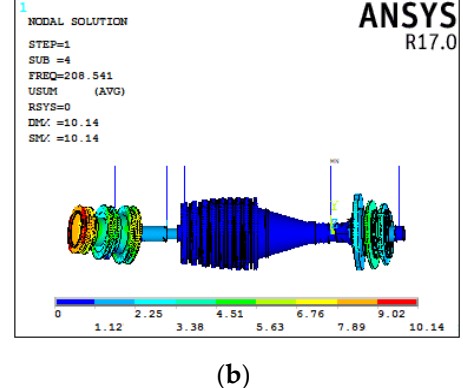

**Figure 23.** Comparison of the 3rd modal between parallel non-concentricity and the front deflection angle non-concentricity: (**a**) δ = 2 mm; (**b**) θ = 0.18°.

Comparing the front deflection angle non-concentricity and the rear deflection angle non-concentricity, it is found that the rear deflection angle non-concentricity has less influence on the vibration characteristics. The main reason for this conclusion is that the blade quality and moment of inertia of the high-pressure turbine are larger than those of the turbofan, which leads to the vibration characteristics of the dual-rotor system being prone to change when the turbine position varies. At the same time, the intermediate bearing of the dual-rotor system is close to the rear support of the high-pressure impeller. The front deflection angle non-concentricity is more severely affected by the intermediate bearing than the rear deflection angle non-concentricity. In short, the main reason is that the position of the intermediate bearing, the mass of the blade and the moment of inertia affect the natural frequency and modal of the dual-rotor.

In summary, during the assembly process of the dual-rotor system of the engine, the parallel non-concentricity and the front deflection angle non-concentricity of the dual-rotor should be controlled through research and comparison. Because the 2nd and 3rd order modal interchanges earlier than the 4th and 5th order modal interchanges, the non-concentricity should be controlled as minutely as possible to prevent the occurrence of the 2nd modal and 3rd order modal interchange, in order to avoid a bending modal appearing in advance. The parallel non-concentricity should be controlled within 2 mm, and the front deflection angle non-concentricity should be controlled within $0.18°$. Therefore, it is ensured that the vibration characteristics of the dual-rotor system of this aero engine do not change significantly.

## 6. Conclusions

In this paper, a three-dimensional finite element solid model of an aero-engine dual-rotor system has been established. Three different modes of parallel non-concentricity, front deflection angle non-concentricity and rear deflection angle non-concentricity have been simulated on the solid finite element model. The influence of three different heart modes on the vibration characteristics of the system has discussed in detail. The main conclusions are as follows.

(1)  Parallel non-concentricity and front deflection angle non-concentricity have a significant impact on the vibration characteristics of the dual-rotor system, and the influence law is similar, while the back-deflection angle non-concentricity has little effect on the vibration characteristics of the dual-rotor system. Therefore, these two types disagreement requires focus control.

(2)  With the increase in the parallel non-concentricity and the front deflection angle non-concentricity, the natural frequency of the bending mode of the dual-rotor system will gradually decrease, while the natural frequency of the local mode will not change much. When the parallel non-concentricity reaches 2 mm or when the front deflection angle non-concentricity reaches $0.18°$, the 2nd order mode and the 3rd order mode are interchanged. When the parallel non-concentricity reaches 2.25 mm or the front deflection angle non-concentricity reaches $0.22°$, the 4th order mode and the 5 order mode will be interchanged. The phenomenon of modal interchange will make the bending mode of the engine appear in advance, causing the engine to deviate from the design working condition.

(3)  The bending modes of the high-pressure and low-pressure rotors will mutually transform in terms of the modal changes of the parallel non-concentricity and the front deflection angle non-concentricity. Moreover, the new bending method deforms more and causes more damage to the rotor.

(4)  Parallel non-concentricity and front deflection angle non-concentricity need to be controlled because the 2nd order and 3rd order modes are interchanged earlier than the 4th order and 5th order modes are interchanged. In order to prevent the occurrence of mode switching, it is recommended that the parallel non-concentricity be controlled within 2.0 mm, and the front deflection angle non-concentricity should be controlled within $0.18°$ of the non-concentricity.

**Author Contributions:** Conceptualization, L.H.; methodology, S.H. and L.H.; investigation, S.H.; data analysis, S.D. and Y.C. (Yufeng Cai); writing—original draft preparation, S.H.; writing—review and editing, L.H. and Y.Y.; supervision, Y.C. (Yushu Chen); funding acquisition, L.H. All authors have read and agreed to the published version of the manuscript.

**Funding:** This research was funded by National Major Science and Technology Projects of China, grant number 2017-IV-0008-0045, National Natural Science Foundation of China, grant numbers 11972129 and 12172307, and the Opening Project of Applied Mechanics and Structure Safety Key Laboratory of Sichuan Province, grant number SZDKF-201903.

**Institutional Review Board Statement:** The study was conducted according to the guidelines of the Declaration of Helsinki, and approved by the Institutional Review Board.

**Data Availability Statement:** All data included in this study are available upon request by contact with the corresponding author.

**Acknowledgments:** The authors are very grateful for help of editors and reviewers.

**Conflicts of Interest:** The authors declare no conflict of interest.

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
