# Peer review of "Vibration Characteristics of a Dual-Rotor System with Non-Concentricity"

_machines, doi:10.3390/machines9110251_

Round 1

Reviewer 1 Report

In the present paper, the dynamics of an aero-engine dual-rotor system is investigated by means of experiments and 3D-FEM analyses. Three different modes related to the dual-rotor non-concentricity induced by the assembly process are identified, i.e., parallel, front deflection angle and rear deflection angle. The influence of the three different selected non-concentricity modes on the natural frequencies of the considered system is analysed.

The paper is well written and organised, presenting several FEM analyses and reporting interesting observations. The dual-rotor system is theoretically modelled and experimentally verified in proper form. The three non-concentricity modes are correctly defined. Moreover, the effect of non-concentricity on dual-rotor system natural frequencies, mode shapes and critical speed is correctly investigated.

In order to consider this paper as acceptable for publication in Machines, the Authors are invited to carefully consider the following notes in the revised version of the work.

1) There are many sentences in the English language that are not technically correct. For example, in the title the phrase "natural vibration characteristics" should be replaced with "vibration characteristics", within which natural frequencies and mode shapes are usually considered (also damping, but not in this paper). The same modification would be made in the Abstract, Introduction and in many parts of the text. As a work on the dynamics of a mechanical system is carried out in this paper, it is very important to use universally accepted terminology as correct for the process parameters.

2) Quality of Figures 4, 8, 16 should be deeply improved (please, also enlarge these figures).

3) Caption of Table 1 is not correct and must be rewritten. As an example, it could become "Comparisons between experiments and FEM analyses".

4) The meaning of "error" in Table 1 is clear. Instead, it is necessary to better explain what "error" means in Tables 3 onwards.

Therefore, by considering the previous notes, in the opinion of the Reviewer the present paper should be accepted for publication by minor revision.

Author Response

1)    There are many sentences in the English language that are not technically correct. For example, in the title the phrase "natural vibration characteristics" should be replaced with "vibration characteristics", within which natural frequencies and mode shapes are usually considered (also damping, but not in this paper). The same modification would be made in the Abstract, Introduction and in many parts of the text. As a work on the dynamics of a mechanical system is carried out in this paper, it is very important to use universally accepted terminology as correct for the process parameters.

Answer: The errors in the article have been corrected and more professional terms have been applied. In particular, the issue of "natural vibration characteristics" proposed by the author was replaced with "vibration characteristics". Similar related issues have been revised in the abstract, introduction and main text, and have been marked in the text.

2)    Quality of Figures 4, 8, 16 should be deeply improved (please, also enlarge these figures).

Answer: Figures 4, 8, and 16 have been revised to improve the clarity of the pictures. These pictures have been enlarged.

3)    Caption of Table 1 is not correct and must be rewritten. As an example, it could become "Comparisons between experiments and FEM analyses".

Answer: Caption of Table 1 has been rewritten as "Natural frequencies comparisons between experiments and FEM analyses". After re-checking, we found that some other titles also have this problem. We have changed them.

4)    The meaning of "error" in Table 1 is clear. Instead, it is necessary to better explain what "error" means in Tables 3 onwards.

Answer: We have explained the meaning of the errors in Table 3, and made changes and explanations in the text.

Reviewer 2 Report

This paper can definitively not be published in its current state!

The English is as bad that it is hard to get what the author wants to communicate. This must be massively revised! Even things like "vibration modes" are constantly wrong.

Also there are a lot of very obvious format errors, which I just see directly by looking on it. Obviously, the authors do not care about a proper use of the format template. This is disappointing.

Here are my further comments:

  • Would be interesting to say a bit more about the engine, the weight for example. Would be nice to get an better idea of it.
  • Pictures of the real engine should be added.
  • The authors cite almost exclusively Chinese literature sources. That is not acceptable. I do not believe that nobody else on earth is dealing with this topic.
  • It is unclear, how the experiments are done. The experimental setup must be described in detail and pictures must be added.
  • It also unclear, how the problem is modeled. How is the assembly considered in the simulation. Which type of contacts are used. How are the different misalignment cases are modeled.
  • It is also not clear, how the different misalignment cases are realised for the experimental investigations.
  • The authors observe an error of 17% and just conclude the model is good. That is not sufficient. They have to investigate how the model could be adapted to achieve a better agreement and also to figure out what are most important influence factors.
  • The pictures are often to small and the quality is too bad (pixelated and blurred)
  • Fig 6 is identical to Fig 5, obviously a wrong picture was pasted?!

Author Response

1)    Would be interesting to say a bit more about the engine, the weight for example. Would be nice to get a better idea of it.

Answer: Thanks for the reviewer’s comment, the modeling data are taken from an experimental rigid which has part of the structure of a real engine, and the natural frequency and modal tests are conducted on the experimental rigid to verify the finite element model. The picture of the engine has been added in section 2.2 of the revised version of manuscript.

2)    The authors cite almost exclusively Chinese literature sources. That is not acceptable. I do not believe that nobody else on earth is dealing with this topic.

Answer:We have added some references in the revised version of manuscript.

3)    It is unclear, how the experiments are done. The experimental setup must be described in detail and pictures must be added.

Answer: The natural frequency and modal test of the engine is supplemented in the article, including the equipment, scheme, process and results of the test, which are described in the article. The specific location is in section 2.2 of the revised version of manuscript.

4)  It also unclear, how the problem is modeled. How is the assembly considered in the simulation? Which type of contacts are used. How are the different misalignment cases are modeled?

Answer:We have explained the modeling process and described the realization process of different heart models. The specific location is as follows:

2.1. The model of the dual-rotor system

The actual structure is simplified, the bolt connection and coupling connection are simplified into adhesive structure; the position of tenon, chamfer and comb disc are equivalent; the blade is treated by the method of equivalent mass center, and the solid finite element model is established

3.3  Realization of the non-concentricity in the dual-rotor system model.

In the finite element model, the high-pressure rotor is moved upward to produce parallel decentraction, forming a model of parallel non-concentricity. By controlling the displacement of the high-pressure rotor, the parallel non-concentricity of two axes is realized. In order to make the front deflection angle non-concentricity, the rear support of the high-pressure rotor is shifted upward to form the front deflection angle model of the dual-rotor. The front deflection angle non-concentricity is shifted upward by controlling the rear support of the high-pressure rotor. The front support of the high-pressure rotor is deflected upward to make the rear deflection angle, forming a model of the rear deflection angle non-concentricity.

5)  It is also not clear, how the different misalignment cases are realised for the experimental investigations.

Answer:Thanks for the reviewer’s comment. In this paper we mainly focus on the simulation of the non-concentricity failure based on the finite element model. The model is examined by the modal test. We will carry out experiments with non-concentricity in future studies by using a simplified experimental rigid.

6)  The authors observe an error of 17% and just conclude the model is good. That is not sufficient. They have to investigate how the model could be adapted to achieve a better agreement and also to figure out what are most important influence factors.

Answer: We are sorry to make the confusion of understanding. The basis for judging the quality of the model in the article is not the error of 17% but the occurrence of modes exchange. When the exchange of modes occurs, the error of the natural frequency of the non-concentric model will reach 17%. The corresponding explanation is in section 5.2 of the paper.

7)  The pictures are often to small and the quality is too bad (pixelated and blurred),Fig 6 is identical to Fig 5, obviously a wrong picture was pasted?!

Answer: Thanks for the reviewer’s comment. We have carefully corrected some picture mistakes in the article, the duplicate problems in picture 5 and picture 6 are solved.